# SYNCHRONOUS SCENE TEXT SPOTTING AND TRANSLATING

## ABSTRACT

Text image machine translation aims to translate the content of textual regions in images from a source language to a target language. Compared with traditional document, images captured in natural scenes have more diverse text and more complex layout, posing challenges in recognizing text content and predicting reading order within each text region. Current methods mainly adopt pipeline pattern, in which models for text spotting and translating are trained separately. In this pattern, translation performance is affected by propagation of mispredicted reading order and text recognition errors. In this paper, we propose a scene text image machine translation approach by implementation of synchronous text spotting and translating. A bridge and fusion module is introduced to make better use of multi-modal feature. Besides, we create datasets for both Chinese-to-English and English-to-Chinese image translation. Experimental results substantiate that our method achieves state-of-the-art translation performance in scene text field, proving the effectiveness of joint learning and multi-modal feature fusion.

## 1 INTRODUCTION

In recent times, there has been a notable rise in interest surrounding research on Text Image Machine Translation(TIMT) (Mansimov et al., 2020; Hinami et al., 2021; Ma et al., 2022; Tian et al., 2023; Zhu et al., 2023; Lan et al., 2023; Ma et al., 2024; Lan et al., 2024; Qian et al., 2024). TIMT aims at identifying one or multiple text regions within an image and translating the textual content from a source language to a target language. A text region is an area in the image containing several words ordered based on spatial and semantic consistency, also called as paragraph from the aspect of natural language. Correct reading order means that the text content in each isolated region follows the semantic logic of natural language and transmits a complete message (Xue et al., 2022). TIMT methods could be classified into two patterns, pipeline and synchronous. Pipeline methods achieve TIMT in multi-steps, including detecting the position or border of the text region, recognizing its text content in the source language and translating it into the target language. Among them, detection and recognition can also be end-to-end and referred to as text spotting. In pipeline methods, text spotting model and translating model are trained separately. On the opposite, synchronous methods unify the two models by jointly learning with multi-tasks. Unlike pipeline, intermediate recognition result is unnecessary in synchronous methods, which is end-to-end, but added to multi-tasks training for the potentially benefit of enhancing translation performance (Ma et al., 2022).

The majority of existing TIMT methods adopt pipeline pattern. However, joint learning demonstrates numerous advantages compared with separate one. Not only does it reduce the propagating error of text recognition, but also makes better use of multi-modal feature. For scene text image machine translation(Scene TIMT), the problem of error propagation becomes more significant due to the variety of image backgrounds and font irregularities. The wide range of visual complexities in scene text images makes it harder for text spotting model to achieve high accuracy and reliability (Long et al., 2021). Consequently, the negative impact of these errors is accumulated and amplified in the translation process (Lan et al., 2023). Furthermore, the intricacy and diversity of layouts in scene text images can lead to incorrect predictions of reading order, thereby further degrading the performance of translation. To address these challenges, we propose a synchronous approach by jointly training text spotting and translating models. The difference of translation result between existing pipeline methods and our proposed approach are illustrated in Figure 1 by an example of Scene TIMT. The layout of text in this example does not satisfy the common zig-zag shaped scanning order and the art font of Chinese characters makes it hard to recognize text content of the

source language correctly. The pipeline result exhibits two recognition errors, and its predicted reading order, as indicated by the yellow arrows in the figure, is inaccurate. This inaccuracy results in erroneous translations on the first half of the sentence. Conversely, our synchronous method predicts correct reading order and fewer recognition errors, and the usage of additional visual information ensures that the translation output remains unaffected by such issues.

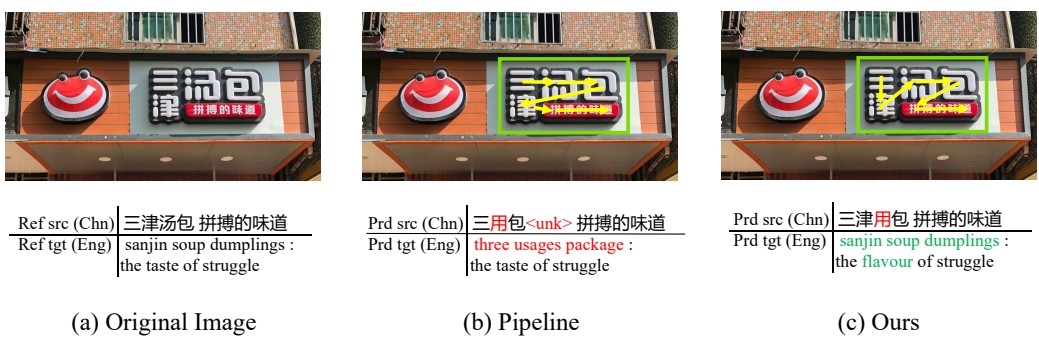

| | |
|---|---|
| Ref src (Chn) | 三津汤包 拼搏的味道 |
| Ref tgt (Eng) | sanjin soup dumplings : the taste of struggle |

| | |
|---|---|
| Prd src (Chn) | 三用包 <unk> 拼搏的味道 |
| Prd tgt (Eng) | three usages package : the taste of struggle |

| | |
|---|---|
| Prd src (Chn) | 三津用包 拼搏的味道 |
| Prd tgt (Eng) | sanjin soup dumplings : the flavour of struggle |

(a) Original Image      (b) Pipeline      (c) Ours

Figure 1: An Example of Scene TIMT to Show Difference Between Pipeline and Ours

Our contributions in this paper are summarized as follows:

- We propose a synchronous text spotting and translating approach for Scene TIMT by jointly training our model with three sub-tasks, detecting text regions in the image, recognizing the source language text content and translating it into the target language.
- A Bridge and Fusion(BAF) module is designed to connect and fuse visual feature with textual feature and enhance translation performance.
- We create Scene TIMT datasets for both Chinese-to-English and English-to-Chinese translation. Label includes text regions' coordinates and their matched bilingual sentence pairs. Words in each region are arranged in semantic reading order.
- Experimental results show that our method achieves the state-of-the-art performance and prove the advantages of joint learning. Besides, ablation study on multi-modal feature fusion demonstrates that BAF module further improves the model's translation ability.

## 2 RELATED WORK

### 2.1 TEXT SPOTTING

**Arbitrary Shaped Text Region** Optimizing for arbitrary shaped text region is one of the popular topics in text spotting field. Studies mainly focus on finding a more suitable format to represent the text region (Liu et al., 2020; 2021a; Peng et al., 2022; Liu et al., 2023; Kil et al., 2023; Wang et al., 2020; Tang et al., 2022). Liu et al. (2020; 2021a) import Bezier control points coordinate for curved text, and Peng et al. (2022); Liu et al. (2023) adopt single-point coordinate to cover arbitrary shaped area. Kil et al. (2023) applies multi-format coordinate by adding a coordinate prompt. Besides, detection granularity represents another research direction. Baek et al. (2019; 2020) introduce character-level attention to achieve fine-grained text spotting and improve the detection flexibility.

**Joint Learning and Multi-task Training** Owing to advancements in computational resources, recent studies have increasingly focused on enhancing performance by simultaneously training the detector and recognizer (Feng et al., 2019; Baek et al., 2020; Liu et al., 2020; 2021a; Qiao et al., 2021; Wang et al., 2021; Zhang et al., 2022; Peng et al., 2022; Liu et al., 2023; Huang et al., 2022; 2023; Kittenplon et al., 2022).Joint learning optimizes detection and recognition in unified architectures (Huang et al., 2022; 2023). Moreover, multiple decoders is a trend for multi-task training. Zhang et al. (2022) designs two parallel decoders, in which one is for recognition, another is for generating fine-grained coordinate . Kittenplon et al. (2022) adds a third branch for text segmentation. Multi-task training could enrich the information contained in visual feature.

## 2.2 Multi-modal Translation

Multi-modal Translation(MMT) is based on text-only machine translation, while the latter is a relatively stable technology. An additional visual model is needed for image feature extraction, and MMT achieves fusion of visual-textual feature by attention mechanism (Calixto et al., 2017; Elliott & Kádár, 2017; Wu et al., 2021; Li et al., 2022a; Lan et al., 2023). According to input image's content, MMT methods could be classified into two types, textual image and non-textual image.

**Textual Image** MMT with textual image as input translates text content with the assistance of semantic information derived from the visual feature map. This semantic information has a absolute correlation with text regions in the image. Lan et al. (2023) builds a code book in training process by clustering the visual feature vectors with similar semantic. In inference process, the code book feature which is closest to current image will be chosen to construct multi-modal feature for translation. The translation performance of MMT further improves based on the text-only machine translation due to the introduction of visual information. However, this method relies on the external Optical Character Recognition(OCR) (Mori et al., 1992) tool. Another way of utilizing visual feature is feeding it directly into the translation model and achieving end-to-end image translation (Mansimov et al., 2020; Ma et al., 2022; 2024; Lan et al., 2024). These models are trained to deal with only simple layout cases with a single paragraph and scanning reading order, so they are not able to deal with complex scene text images.

**Non-textual Image** MMT with non-textual image as input is based on the assumption that the source language text is already given correctly. The optimization methods in this filed do research on making use of visual feature as additional information to improve traditional text-only machine translation. One research direction focuses on exploring model structure for better visual-textual feature fusion: Wu et al. (2021); Li et al. (2022a) use gated vector; Li et al. (2022a) imports selective attention mechanism; Ye et al. (2022) adopts interactive fusion with cross-modal relation-aware mask mechanism. Another research direction is optimizing training process. Cheng et al. (2024) limits the textual input and guides model to pay more attention to visual information. Guo et al. (2023); Cheng et al. (2023) focus on the problem of insufficient text-image pairs and consider synthetic data generation. Guo et al. (2023) uses consistency training to reduce the difference between the output of the transformer decoder for synthetic and real data, and Cheng et al. (2023) applies asymmetric contrastive learning to mitigate the negative impact of noise in those generated pairs.

## 3 Methodology

### 3.1 Overview

The input of Scene TIMT is an image $I \in \mathbb{Z}^{H \times W \times 3}$. Information of one text region contains its coordinate, source language text and target language text for translation:

- Scene text regions exhibit a wide range of diversity in shape and size. To accommodate various scenarios, we employ multi-format coordinates of three types: boxes, quadrilaterals, and polygons with sixteen vertices. They are defined by a type prompt followed with a fixed number of 2-dimensional vertices, $\{\langle box \rangle, x_0, y_0, x_1, y_1\}$, $\{\langle quad \rangle, x_0, y_0, x_1, y_1, x_2, y_2, x_3, y_3\}$ or $\{\langle poly \rangle, x_0, y_0, ..., x_{15}, y_{15}\}$. Single point coordinate is not used as we need explicit region border in prediction output to retain the capability of rendering the translated text back onto the image.

- Both source and target language text use Byte Pair Encoding (Bostrom & Durrett, 2020), and they share same vocabulary with size 60K. All texts are described as padded token sequence with fixed length, like $\{s_0, s_1, \langle pad \rangle, ..., \langle pad \rangle\}$ and $\{t_0, t_1, \langle pad \rangle, ..., \langle pad \rangle\}$.

The overview of our proposed method is shown in Figure 2. It includes four main modules, Visual Feature Extraction, Detection & Recognition, Bridge & Fusion and Translation. Among them, Visual Feature Extraction and Detection & Recognition construct a text spotter, which is based on UNITS (Kil et al., 2023), a sequence generation model for end-to-end text spotting. Translation module is based on Transformer (Vaswani, 2017).

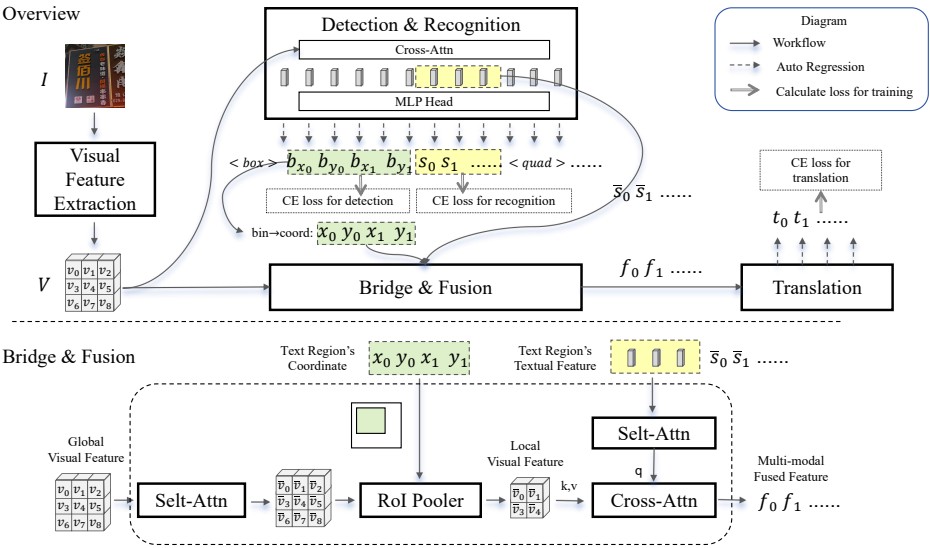

Figure 2: The Overview of Our Proposed Method and Details of BAF Module

## 3.2 VISUAL FEATURE EXTRACTION

The backbone for visual feature extraction is Swin-Transformer (Liu et al., 2021b), pre-trained with computer vision tasks[1]. To capture high-level visual information, we select the output of the last and also deepest layer as visual feature, $V = \{v_0, v_1, ..., v_m\}$. Visual feature could be used in both 1-dimensional sequence format and 2-dimensional map format. $m = h \times w$ is the length in sequence format, where $h$ and $w$ are the height and width of map format, scaled by 1/32 from the original image size $H$ and $W$. The coordinate is based on visual feature map, so it is correspondingly scaled by 1/32 from the original coordinate in the image.

## 3.3 DETECTION & RECOGNITION

The auto-regressive structure for sequence generation has been proved effective in end-to-end text spotting task previously (Kil et al., 2023). Similarly, Detection & Recognition module is constructed by eight stacked Transformer Decoder (Vaswani, 2017) layers. The input for these decoders is the visual feature $V$. Information of object text regions are generated in the format of a coordinate-text-mixed sequence. Since the model is trained with paragraph-level label (not word-level) of text regions, layout analysis are implicitly embedded in this sequence generation process, including paragraph segmentation and reading order prediction. The recognition output is $S = \{s_0, s_1, ...\}$. For detection output, the coordinate like $\{x_0, y_0, x_1, y_1\}$ is converted into discrete bins $B = \{b_{x_0}, b_{y_0}, b_{x_1}, b_{y_1}\}$ in order to integrate detection and recognition into a unified classification task paradigm (Kil et al., 2023). The conversion method is Equation 1, where $n_{bins} = 1000$ is the number of bins. For the $n$-th vertex of the coordinate, $x_n$ is the position along width axis in visual feature map, and $y_n$ is the position along height axis:

$$b_{x_n} = \lceil x_n/w \times n_{bins} \rceil$$
$$b_{y_n} = \lceil y_n/h \times n_{bins} \rceil \tag{1}$$

## 3.4 BRIDGE & FUSION

Bridge & Fusion module is shown in the lower half of Figure 2. The input visual feature $V$ is globally extracted from the whole image. The input textual feature $\{\bar{s}_0, \bar{s}_1, ...\}$ is the embedding

---

[1]`https://github.com/SwinTransformer/storage/releases/download/v1.0.0/`
`swin_base_patch4_window7_224_22k.pth`

sequence generated from the last decoder layer in Detection & Recognition module, matched with recognized text $S$. Both visual and textual feature are applied with a self-attention layer, which means that visual feature is able to collect global visual information which is helpful for translation task before cropping local feature for the specific text region. Coordinate of each text region like $\{x_0, y_0, x_1, y_1\}$, which is converted from the discrete bins $B$ generated in Bridge & Fusion Module, is fed into Region of Interest (RoI) Pooler and guides the pooler to crop local visual feature for each text region from the whole visual feature map. These self-attention and cropping operations make a bridge for connecting features extracted from different modules. Then the processed textual feature is fed into cross-attention as query, while the cropped local visual feature as key and value. The output multi-model fused feature $F = \{f_0, f_1, ...\}$ has same length as recognized source language text $S$, but combined with more details of visual information. Notably, the textual feature is concatenated with two position embedding before fed into fusion module as query. One is 2-dimensional and represents the global position of each feature vector in the feature map of the whole image, another is 1-dimensional and represents the local position of each feature vector in the recognized text sequence. That way, even if the recognition result is wrong, BAF module is still able to guide cross-attention layer to collect visual information for translation according to position-related part in the query.

## 3.5 TRANSLATION

Translation module is based on typical machine translation model, Transformer (Vaswani, 2017). This module regressively generates translation result $T = \{t_0, t_1, ...\}$ based on the multi-model fused feature $F$ from BAF module. We apply beam search in decoding process(beam size is 5).

## 3.6 OPTIMIZATION OBJECT

Scene TIMT is optimized with multi-tasks. The final loss is the weighted sum of three sub-losses, respectively for detection, recognition and translation. These three sub-tasks are all classification tasks, so their losses are calculated with Cross-Entropy(CE). The loss function is shown in Equation 2, where hat mark means ground true label. The $\alpha$ is the sum weight, which is not sensitive in training and could be set as 0.1, 0.5 or 0.9.

$$loss = \alpha \times (CE(B, \hat{B}) + CE(S, \hat{S})) + (1 - \alpha) \times CE(T, \hat{T}) \tag{2}$$

# 4 EXPERIMENT

## 4.1 DATASETS

We create Scene Text Spotting and Translation 800,000 (STST800K) dataset, which contains both synthesized and real data for Chinese-to-English and English-to-Chinese translation tasks. The overview is in Table 1, and example cases are shown in Figure 3.

### 4.1.1 DATA SYNTHESIZING

Image source is COCO (Lin et al., 2014) with more than 330K images constructed for scene understanding tasks. We use PaddleOCR[2] tool to exclude images with text content. After filtering, there are nearly 100K images left as synthesizing background material. We extract the depth mask[3] for each image and generate its segmentation mask[4] as well. With these two masks, images are segmented into several regions and some of them are chosen to render text of the source language for their suitable shape and sufficient area. The source-target language text pairs are from Machine Translation Challenge WMT22 (Kocmi et al., 2022). There are nearly 55M Chinese-English pairs in the zh-en subset of WMT22. After deleting long text which is not suitable for rendering on image,

---

[2]https://github.com/PaddlePaddle/PaddleOCR
[3]https://bitbucket.org/fayao/dcnf-fcsp/src/master/
[4]https://github.com/jponttuset/mcg

| Chinese-to-English | Count | English-to-Chinese | Count |
|---|---|---|---|
| SynthChn | 500,000 | SynthEng | 300,000 |
| OCRMT30K (Lan et al., 2023) | 30,000 | HierText (Long et al., 2022) | 10,000 |
| ReCTS (Liu et al., 2019) | 20,000 | CTW1500 (Xue et al., 2022) | 1,500 |

Table 1: Overview of STST800K

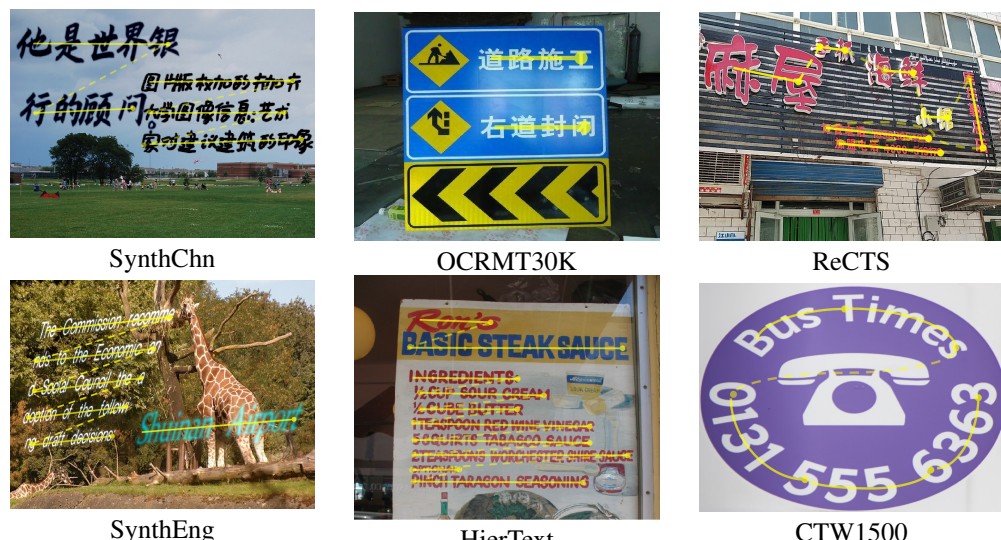

Figure 3: Example Cases of STST800K (Yellow Arrow Presents Reading Order)

there are about 39M pairs left. We use the filtered images and source language text to synthesize Scene TIMT data. The synthesizing code is modified from SynthText[5].

### 4.1.2 REAL DATA RELABELING

OCRMT30K (Lan et al., 2023) has complete annotations for Scene TIMT, including paragraph-level coordinate, recognition result and its matched translation. ReCTS (Liu et al., 2019)'s label is word-level and does not have translation. HierText (Long et al., 2022) is a hierarchical labeled scene text dataset, and its labeling structure is paragraph-lines-words. The words within a line are arranged in reading order, however the lines within a paragraph are not. Contextual CTW1500(CTW1500) (Xue et al., 2022)'s label is also word-level but provides reading order information to construct text paragraph from words. Besides, neither HierText nor CTW1500 has translation label. We manually label the text regions and reading order information for test subset of ReCTS and HierText. Other data's reading order and all translation labeling work is achieved by large-scale language model API Tongyi Qianwen[6] and then proofread by humans.

### 4.2 EVALUATION METHOD

Object detection harmonic mean (Hmean), also called F1-score, is used to measure the performance of text region detection, while SacreBLEU (Post, 2018) and COMET (Guerreiro et al., 2023) is for evaluating the quality of translation.

**Hmean** is a comman object detection evaluation method. In our experiments, if the ratio of the intersection area to the union area between predicted coordinate and ground true coordinate is greater than 0.5, this sample is seen as detected correctly.

**SacreBLEU** and **COMET** serve as tools for evaluating the quality of machine translation by assessing the similarity between the generated text and the reference translation label. In our experiments,

---

[5]https://github.com/ankush-me/SynthText
[6]https://tongyi.aliyun.com/

the language model used to calculate COMET is XCOMET-XL[7]. These two scores provide a comprehensive evaluation that encapsulates translation accuracy, fluency, and similarity.

## 4.3 EXPERIMENTAL SETTINGS

Training process includes three steps:

- Pre-train Visual Feature Extraction and Detection & Recognition modules on low resolution $768 \times 768$ with all image data. Pre-train Translation module with text-only data WMT22 (Kocmi et al., 2022). This is for quicker converging.

- Jointly train all modules and improve image resolution from low to high $1920 \times 1920$. Except BAF module, other modules' weights are initialized with pre-trained weights got in the first step. Text-only data is still used in for training to avoid the decreasing of generalization ability in Translation module.

- Fine-tune on target dataset.

Optimizer is AdamW (Loshchilov et al.). For the three steps, batch size = {32, 16, 16}, learning rate = {3e-4, 1.2e-4, 3e-5}. The total number of training batches is 1000K for the first two steps and 300K for the third step. All experiments are done on eight A6000 GPUs.

## 4.4 RESULTS

### 4.4.1 BASELINES COMPARISON

| Pipeline or End-to-End | Method | Model Parameters | GT Coord | | | | Prd Coord | | | | | |
| | | | OCRMT30K | | ReCTS | | OCRMT30K | | | ReCTS | | |
| | | | B | C | B | C | H | B | C | H | B | C |
| Pipeline (PaddleOCR + Multimodal Machine Translation) | Gated Fusion (Wu et al., 2021) | 96M | 29.87 | 75.60 | 24.31 | 66.09 | - | - | - | - | - | - |
| | Selective Attn (Li et al., 2022a) | 96M | 26.80 | 70.69 | 21.24 | 66.83 | - | - | - | - | - | - |
| | VALHALLA (Li et al., 2022b) | 260M | 28.12 | 70.02 | 25.68 | 69.11 | - | - | - | - | - | - |
| | E2E-TIT (Ma et al., 2022) | 137M | 16.30 | 42.88 | 10.16 | 39.10 | - | - | - | - | - | - |
| | MCTIT (Lan et al., 2023) | 158M | 31.07 | 80.93 | 24.42 | 69.79 | - | - | - | - | - | - |
| End-to-End (Text Detection & Translation) | ABCNetv2 (Liu et al., 2021a) | 51M | 2.17 | 33.69 | 4.34 | 16.14 | 52.55 | 3.78 | 37.43 | 54.59 | 6.10 | 25.95 |
| | SWINTS (Huang et al., 2022) | 177M | 16.17 | 39.60 | 18.84 | 45.60 | 69.73 | 16.80 | 42.00 | 66.37 | 18.40 | 44.53 |
| | SPTSv2 (Liu et al., 2023) | 36M | 1.90 | 38.32 | 8.50 | 36.80 | 51.14 | 5.47 | 34.92 | 59.42 | 12.98 | 37.45 |
| | UNITS (Kil et al., 2023) | 133M | 24.06 | 66.34 | 22.45 | 58.31 | **74.24** | 23.64 | 60.32 | **71.79** | 20.73 | 50.58 |
| Synchronous | Ours | 288M | **35.60** | **84.46** | **28.23** | **75.55** | 65.15 | **37.51** | **85.38** | 65.26 | **25.55** | **71.52** |

Table 2: Comparisons with State-of-the-arts on Chinese-to-English Datasets.

| Pipeline or End-to-End | Method | Model Parameters | GT Coord | | | | Prd Coord | | | | | |
| | | | HierText | | CTW1500 | | HierText | | | CTW1500 | | |
| | | | B | C | B | C | H | B | C | H | B | C |
| Pipeline | MCTIT (Lan et al., 2023) | 158M | 28.00 | 72.60 | 27.45 | 73.38 | - | - | - | - | - | - |
| End-to-End | ABCNetv2 (Liu et al., 2021a) | 51M | 6.17 | 40.80 | 3.30 | 22.53 | 33.60 | 3.55 | 27.34 | 36.69 | 4.40 | 22.93 |
| | SPTSv2 (Liu et al., 2023) | 36M | 4.09 | 29.32 | 6.50 | 33.02 | 48.14 | 5.40 | 36.77 | 40.40 | 18.22 | 44.76 |
| | UNITS (Kil et al., 2023) | 133M | 22.02 | 58.10 | 21.80 | 60.37 | **55.48** | 19.58 | 58.54 | **78.79** | 22.00 | 51.74 |
| Synchronous | Ours | 288M | **30.80** | **72.63** | **34.50** | **75.60** | 54.40 | **35.00** | **78.98** | 72.28 | **36.76** | **76.90** |

Table 3: Comparison with State-of-the-arts on English-to-Chinese Datasets

To evaluate our proposed method comprehensively, we select two kinds of existing methods as baselines, pipeline and end-to-end:

- **Pipeline Baselines** These methods are MMT model using external OCR tool to do text spotting or assuming that the recognition result is given. However, the text region detected by OCR tool might be smaller than the ground truth, since label for TIMT is paragraph-level, and for scene text, words belonging to the same paragraph might be split into different region by OCR because of not gathered spatially on the image. To solve this problem, we use ground true coordinate(GT Coord) of text region to crop images and then use OCR tool(PaddleOCR) to recognize the text content within the cropped pieces. If there are multiple objects detected by OCR tool in one text region, we arrange them in zig-zag shaped scanning reading order and see them as a whole paragraph when subsequently applying multi-modal translation. Testing with GT Coord ensures that pipeline methods can be

---

[7]https://github.com/Unbabel/COMET

compared with end-to-end methods without considering the factor of text region detection. Moreover, we can observe the effect of reading order error due to the assumption of the simple layout rule. To compare fairly, we also use GT Coord to test end-to-end methods. Besides, end-to-end methods are individually tested with predicted coordinate(Prd Coord), and the purpose of this test is further evaluating the whole performance of end-to-end methods for Scene TIMT task.

- **End-to-End Baselines** Since a comprehensive end-to-end model for synchronous detecting and translating remains elusive in existing methods, we select state-of-the-art text spotting models as end-to-end baselines by training them for translation instead of recognition.

The result of baselines comparison is shown in Table 2 and Table 3. Our proposed method outperforms both pipeline and end-to-end baselines and achieves the state-of-the-art on datasets of both Chinese-to-English (OCRMT30K (Lan et al., 2023) and ReCTS (Liu et al., 2019)) and English-to-Chinese (HierText (Long et al., 2022) and CTW1500 (Xue et al., 2022)).

UNITS (Kil et al., 2023) is the best end-to-end baseline. Visual Feature Extraction module and Detection & Recognition module in our proposed method are based on it. The scores of the best end-to-end baseline are still much lower than our proposed method in both GT Coord column and Prd Coord column, this fact indicates that additional translation module is necessary in Scene TIMT task.

MCTIT (Lan et al., 2023) is the best pipeline baseline, whose performance is better than that of the best end-to-end baseline. To further prove the advantage of joint learning, we conduct an additional test on MCTIT and our proposed method by using ground true coordinate and recognition(GT Coord & Rec) as input. This test evaluates the translation performance individually without considering the factor of OCR. The result in Table 4 shows that our proposed method performs better than the best pipeline baseline even when using the entirely correct result of OCR, especially on ReCTS(Liu et al., 2019), the test set which has more complex layout than OCRMT30K(Lan et al., 2023).

| Method | GT Coord & Rec | | | |
| --- | --- | --- | --- | --- |
| | OCRMT30K | | ReCTS | |
| | B | C | B | C |
| MCTIT (Lan et al., 2023) | **49.07** | 91.34 | 31.69 | 78.54 |
| Ours | 48.63 | **92.15** | **34.04** | **81.37** |

Table 4: Test With Ground Truth OCR Result

### 4.4.2 VISUAL-TEXTUAL LARGE-SCALE LANGUAGE MODEL COMPARISON

Our method focuses on synchronous training of small model. However, it exhibits the largest model size in comparison to the baseline small models. To highlight the advantages of our method, we further compare our model's multi-modal translation ability with visual-textual large-scale language model(VLM). In Table 5, there are three kinds of evaluations. (1) GT Coord + Ours recognition: We ask VLM to translate the recognition result generated by our method (with GT Coord as input). The input for VLM includes the image and the recognition text with a few prompt. (2) GT Coord + VLM recognition: Draw GT Coord on image and ask VLM to translate the text within the given region. (3) Translate Whole Image: Input the original image and ask VLM to translate the whole image. If there are more than one text regions in the image, the translation label for the whole image is a paragraph with all text regions arranged by scanning reading order.

| Method | Parameters | GT Coord + Ours recognition | | | | GT Coord + VLM recognition | | | | Translate Whole Image | | | |
| --- | --- | --- | --- | --- | --- | --- | --- | --- | --- | --- | --- | --- | --- |
| | | OCRMT30K | | ReCTS | | OCRMT30K | | ReCTS | | OCRMT30K | | ReCTS | |
| | | B | C | B | C | B | C | B | C | B | C | B | C |
| Ours | 2.88B | **35.60** | 84.46 | **28.23** | 75.55 | **35.60** | 84.46 | **28.23** | 75.55 | **23.44** | 55.73 | **24.55** | 58.65 |
| Qwen-vl-max (Bai et al., 2023) | - | 21.92 | 83.74 | - | - | - | - | - | - | - | - | - | - |
| mPLUG-OWL2 (Ye et al., 2024) | 8.2B | 12.61 | 78.75 | 7.96 | 68.60 | 0.90 | 61.06 | 0.59 | 53.61 | 0.36 | 28.84 | 1.01 | 32.71 |
| InternVL2-8B (Chen et al., 2024) | 8B | 23.06 | 84.22 | 14.47 | 75.16 | 12.62 | 75.84 | 7.46 | 67.48 | 3.52 | 49.64 | 2.21 | 50.56 |
| TextMonkey (Liu et al., 2024) | 9.7B | 9.69 | 80.85 | 5.78 | 71.00 | 2.60 | 65.46 | 1.61 | 57.80 | 1.13 | 40.36 | 1.28 | 40.75 |
| Qwen2-VL-8B-Instruct (Wang et al., 2024) | 8B | 23.28 | **84.65** | 12.38 | 73.98 | 20.75 | 80.73 | 7.80 | 69.61 | 2.79 | 50.19 | 1.60 | 51.98 |

Table 5: Compare our method with VLM

The result shows VLM has much lower OCR ability than ours, which proves the necessity of small model training in text spotting task. When comparing the performance of multimodal translation,

our model still outperforms or performs comparably to VLM, which highlights the importance of synchronous training. Because Qwen-vl-max does not provide model weight, we could only call server API of Qwen-VL[8], and the evaluation is not applied on all conditions. To further compare our model with Qwen-vl-max, we download the Chinese-to-English image translation dataset of AnyTrans (Qian et al., 2024) and test our model on it. AnyTrans uses Qwen-vl-max to do multi-modal translation. It is noteworthy that we use SacreBLEU (Post, 2018) to calculate BLEU score in this paper, which is the weighted sum of N-gram values of BLEU(N is from 1 to 4), and the calculation of COMET (Guerreiro et al., 2023) calls the latest model XCOMET-XL[9]. When comparing with AnyTrans, we use BLEU score with N=1, and the calculation of COMET score calls the default model wmt-22-comet-da[10]. This is to keep the same configuration as AnyTrans in evaluation process. The result in Table 6 illustrates that our methodology attains comparable performance to AnyTrans in multi-model translation.

| Method | AnyTrans: zh2en | |
|---|---|---|
| | B | C |
| Ours | 47.50 | **79.72** |
| AnyTrans: Qwen-vl-max | **48.70** | 78.00 |

Table 6: Compare our method with AnyTrans

### 4.4.3 ABLATION STUDY

To study the effect of BAF module and multi-modal feature fusion, we set three groups of ablation experiments based on our proposed method to compare with its best performance:

- **Best** Use BAF module and joint learning.
- **Visual Only** Use coordinate of detected text region to crop local feature from the whole visual feature map by RoI Pooler, and feed it into translation model directly without multi-modal fusion.
- **Textual Only** Feed the textual feature from the last layer of Decoder & Recognition module into translation model directly without multi-modal fusion.
- **Remove BAF** Use same model structure as Best for text spotting and translating but train them separately. Translate the recognized result of text spotting directly by translation model. BAF is not used in either the training process or the inference process.

| Ablation Condition | OCRMT30K | | | ReCTS | | |
|---|---|---|---|---|---|---|
| | H | B | C | H | B | C |
| Best | 65.15 | **37.51** | **85.38** | 65.26 | **25.55** | **71.52** |
| Visual Only | 66.06 | 28.24 | 69.80 | 64.93 | 22.33 | 63.02 |
| Textual Only | 62.06 | 30.27 | 75.83 | 59.88 | 20.40 | 55.75 |
| Remove BAF | **71.81** | 31.54 | 78.12 | **68.80** | 22.02 | 61.80 |

Table 7: Ablation Experiment for BAF Module on OCRMT30K and ReCTS Datasets

Table 7 shows that using only visual feature or textual feature is not good as using both of them. Multi-modal fusion could improve the performance of model. Besides, when removing BAF and adopting separate training, the detection score improves to some degree due to the reduction of training tasks in text spotting part, but the translation score drops by one level even if the model structure keeps same. The result of ablation proves the effectiveness of BAF module and multi-model feature fusion.

### 4.4.4 CASE STUDY

Compared with pipeline pattern, joint learning and multi-modal feature fusion can make better use of visual information and reduce the error as well as the effect of error propagating in recognition

---

[8] https://github.com/QwenLM/Qwen-VL
[9] https://huggingface.co/Unbabel/XCOMET-XL
[10] https://huggingface.co/Unbabel/wmt22-comet-da

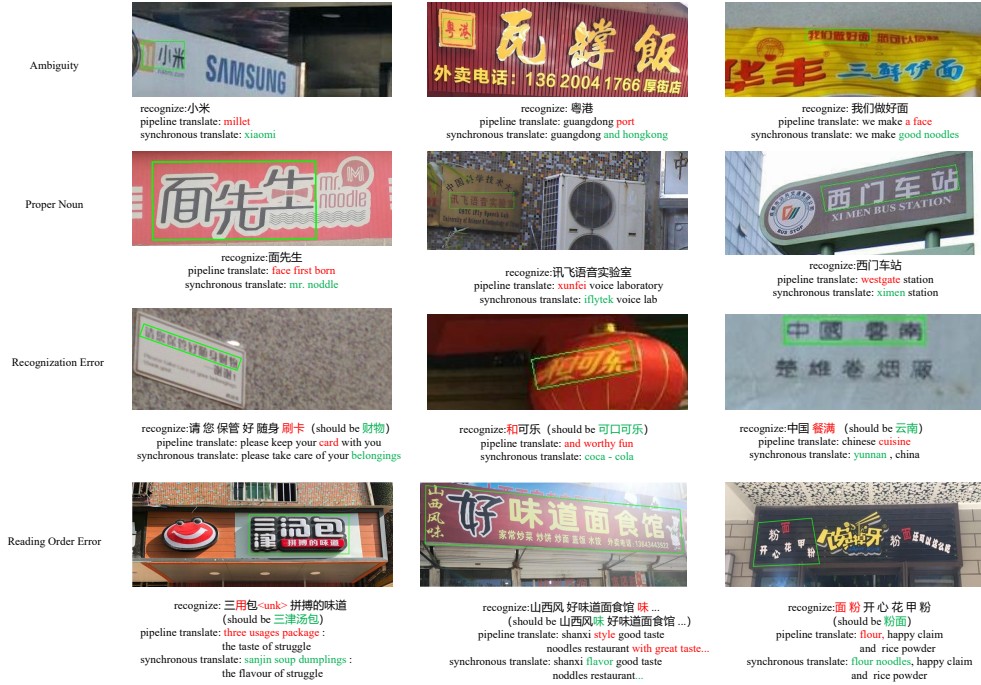

Figure 4: Visualization of Hard Cases for Comparing Pipeline and Synchronous Method

and reading order prediction. To visualize this advantage, we collect three kinds of hard cases in Figure 4 and compare the result of pipeline method and our proposed method:

- **Ambiguity** Same words of the source language might have multiple semantic according to different visual backgrounds. For example "xiaomi" is a trademark on the signboard, so it should not be translated into "millet" though they are matched with same Chinese characters. The translation error caused by such ambiguity in pipeline methods is optimized in the synchronous method.

- **Proper Noun** Sometimes the translation result of target language text exists in the original image, especially for proper noun like address and station name. In pipeline methods, the lack of visual information causes the translation error which could be solved by utilizing multi-modal fusion.

- **Recognition & Reading Order Prediction Error** Joint learning is able to reduce recognition and reading order prediction error in scene text cases with complex layout, resulting in improvement in translation performance. Even the recognition result is wrong, the BAF module in our proposed method is still able to search valid information in visual feature map and generate correct translation result.

## 5 LIMITATION

Our proposed method leverages shared visual feature extracted from the image encoder for multiple tasks, including source language detection, recognition, implicit layout analysis, and target language translation, posing a challenge in balancing the training of these disparate objectives. The shared nature of the visual feature may lead to suboptimal performance for some individual task, as a single set of feature is tasked with serving multiple purposes. Future work could explore extending the visual feature to be multi-way, tailored specifically for each task, as exemplified by Multi-way FFNs (He et al., 2021), potentially enhancing both paragraph detection and translation performance simultaneously.

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
