# OpenReview forum: "Synchronous Scene Text Spotting and Translating"
_ICLR.cc/2025/Conference — Submitted to ICLR 2025_

### Official Review · Reviewer_CjTp · 2024-10-27

**Soundness:** 3
**Presentation:** 3
**Contribution:** 3
**Rating:** 5
**Confidence:** 5

**Summary:**

This paper introduces a unified framework for scene text image spotting and translation. A BAF Module is proposed to connect and integrate visual and textual features. Additionally, the authors have created the Scene TIMT (Scene Text Image Multilingual Translation) datasets for both Chinese-to-English and English-to-Chinese translations. Experimental results demonstrate the effectiveness of the proposed method.

**Strengths:**

1. This paper proposes a unified framework designed to enhance the performance of text image machine translation.
2. This paper introduces a new Scene TIMT (Scene Text Image Multilingual Translation) dataset for this field.

**Weaknesses:**

1. The proposed method involves a larger set of parameters compared to other methods.
2. When comparing with multimodal large models, the analysis lacks a comparison against the most recent large multimodal models, such as mplug-owl (CVPR 24), Monkey (CVPR 24), and InternVL (CVPR 24). It would be beneficial to have the performance results of the large multimodal models after they have been fine-tuned using the corresponding dataset.
3. The authors claim that “In this pattern, translation performance is affected by propagation of mispredicted reading order and text recognition errors.” However, there is a lack of experiments or evidence to verify this claim.
4. Lack of experiment to verify the effectiveness of proposed BAF Module using different text spotter and Translation module.
5. There is a lack of experimental validation to show how a unified framework improves different modules. Does it provide a greater improvement for end-to-end text spotting or for translation?

**Questions:**

Why is it necessary to unify end-to-end spotting and translation? The author did not emphasize the necessity of unification in the paper.

---

> ### Author Response · Authors · 2024-12-02
> **Response to Reviewer CjTp**
>
> We greatly appreciate your valuable time and feedback on our paper. Here are our responses to weakness and questions.
>
> >*“The proposed method involves a larger set of parameters compared to other methods.”*
>
> From Tables 2 and 3, it is evident that our method has a higher number of parameters compared to the baselines.  However, by expanding Table 4 and conducting comparative experiments with large models (thank you for your suggestion to include experiments with large models), we have validated our method achieves the best translation performance in a middle level of model size. For specific results, please refer to the table in the "Performance" section of our General Response 1.
>
> >*“When comparing with multimodal large models, the analysis lacks a comparison against the most recent large multimodal models, such as mplug-owl (CVPR 24), Monkey (CVPR 24), and InternVL (CVPR 24). It would be beneficial to have the performance results of the large multimodal models after they have been fine-tuned using the corresponding dataset.”*
>
> It is indeed beneficial to fine-tune. However, considering the need for a quick performance comparison, we opted to directly test the large models with our data. Since these large models do not undergo a fine-tuning process, we designed three testing scenarios to ensure fairness. Besides, we compare our method with AnyTrans, which is the latest large model method trained specifically on scene TIMT tasks. For dataset fairness, we directly test our model on AnyTrans's dataset without any fine-tuning. And in the future research we will add experiment with model fine-tuning.
>
> For specifics of large model comparing and AnyTrans, please refer to our General Response 2.
>
> >*“The authors claim that “In this pattern, translation performance is affected by propagation of mispredicted reading order and text recognition errors.” However, there is a lack of experiments or evidence to verify this claim.”*
>
> For evidence of propagation's effect, please refer to the experiment result of last raw "Remove BAF" in Table 7 in our paper. Remove BAF means that use text spotter(best one in baselines, UNITS) and translation module in a pipeline way, which does not solve the issue of recognition errors and reading order issues and causes the decreasing in translation performance.
>
> What's more, we could see from the image cases in the third and fourth rows of Figure 4 that there are still recognition error and reading order error in pipeline even if we used the state-of-art text spotter. And for their effect in translation, for example:
> - Figure 4, row 3, column 3: the text in the image was challenging traditional Chinese art fonts, the text spotter mistakenly recognized "雲南" (Yunnan, a province in China) as "餐满" (which can be loosely translated as "full meal"). And the translation "cuisine" is no doubt wrong.
> - Figure 4, row 4, column 3: the complex layout of Chinese characters makes it hard for text spotter to predict correct reading order of "粉" and "面". It wrongly predicted it as "面粉", which was translated into "flour", a basic ingredient. But the correct answer is "粉面", which represents a local food similar to noodles.
>
> >*“Lack of experiment to verify the effectiveness of proposed BAF Module using different text spotter and Translation module.”*
>
> We conducted an ablation experiment on the BAF module to validate its effectiveness. However, to further strengthen the persuasiveness of BAF, it would indeed be a better approach to train various combinations of text spotter and translator while keeping the BAF module fixed. Our subsequent experiments will continue to explore this aspect. Thank you for this valuable suggestion.
>
> >*“There is a lack of experimental validation to show how a unified framework improves different modules. Does it provide a greater improvement for end-to-end text spotting or for translation?”* / *“Why is it necessary to unify end-to-end spotting and translation? The author did not emphasize the necessity of unification in the paper.”*
>
>
> Our study focuses on improving translation performance with unification and multi-task training, as evidenced by comparisons with the best pipeline baseline and the best E2E baseline.
>
> - The best pipeline baseline is MCTIT. To eliminate the impact of the OCR tool, we used ground truth coordinates and recognized text as input, solely comparing translation performance with our method. In Table 4 of the paper, our method demonstrated comparable performance on OCRMT30K and achieved a higher score on ReCTS. Notably, the latter dataset features a complex layout level, as shown in the table in the "Data" section of our General Response 1.
>
> - The best E2E baseline is UNITS. We removed BAF module and test in a pipeline pattern by UNITS text spotter + translation module. The result is the last row of Table 7 of paper. The translation performance(**B**LEU, **C**OMET) of this pipeline is lower than the best row, which uses unification and multi-task training.

---

### Official Review · Reviewer_wQqR · 2024-10-29

**Soundness:** 3
**Presentation:** 2
**Contribution:** 2
**Rating:** 5
**Confidence:** 4

**Summary:**

This paper proposes a synchronous text spotting and translating approach for Scene TIMT by jointly training the model with three sub-tasks, detecting text regions in the image, recognizing the source language text content, and translating it into the target language.
In order to solve the problem of mis-ordered text recognition and translation that occurs in existing pipelined methods, the paper introduces a “Bridging and Fusion (BAF)” module to more effectively utilize visual and textual features to achieve efficient scene text translation. The method is evaluated on the self-constructed STST800K dataset and compared with existing methods.

**Strengths:**

1) The paper proposes that the synchronized training approach can improve the sequential reading accuracy of translation compared to the traditional pipeline model.
2) The proposed BAF module is creative and helpful in fusing visual and textual features.
3) The paper creates a large-scale dataset containing 800,000 samples of Chinese-English translations with real and synthetic data annotations, providing a new benchmark for scenario-based text translation tasks.

**Weaknesses:**

1) For text recognition and translation of complex scenes, especially fine-grained and layout complexity is not reflected in the paper.
2) The way of synchronized training is not clearly described. The model's training data is large, and the resources consumed are not mentioned. Large datasets lead to more resources consumed for training models.

**Questions:**

1) How does the BAF module affect the reliability of the translated output in cases where errors occur in detection or recognition? Could more error analysis be provided to help understand the effects of the fusion of visual and textual features?
2) How about the recognition and translation performance of the model in more complex scenarios?
3) Can the model be applied to translations in languages other than Chinese and English?
4) Can there be a more detailed explanation of the construction of the dataset？

---

> ### Author Response · Authors · 2024-12-02
> **Response to Reviewer wQqR**
>
> We greatly appreciate your valuable time and feedback on our paper. Here are our responses to weakness and questions.
>
> >*“For text recognition and translation of complex scenes, especially fine-grained and layout complexity is not reflected in the paper.”*
>
> **Qualitative explanations** We present an example in Figure 1 to illustrate the issue of reading order, more examples could be found in Figure 3 and Figure 4. The complex layout is presented in aspects like fonts stype, fonts size, text richness and text reading order.
> - For example in Figure 1, the nine Chinese characters use different font styles and sizes, and the reading order is not normal scanning, which could be wrongly detected as "三汤包津" (correct one is "三津汤包", which is a name of a breakfast restaurant). Moreover, "津" could be wrongly recognized as "用" because of art style of font.
> - To understand the complexity of our layout, please see another example in Figure 4-row 4-column 3: the Chinese characters are completely scattered, making it hard for text spotter to predict correct reading order of "粉" and "面". It wrongly predicted it as "面粉", which was translated into "flour", a basic ingredient. But the correct answer is "粉面", which represents a local food similar to noodles
>
> **Quantitative validations** We compare our dataset's layout complexity with the state-of-the-art dataset in scene TIMT field by statistic method. And our dataset shows much more complexity. Please refer to "Data" section in our General Response 1 to see more details.
>
> >*“The way of synchronized training is not clearly described. The model's training data is large, and the resources consumed are not mentioned. Large datasets lead to more resources consumed for training models.”*
>
> The way of synchronized training is described in Chapter 4.3 EXPERIMENTAL SETTINGS, which starts from "Training process includes three steps:...".
>
> We acknowledge that the resources utilized are substantial. This was also noted in Chapter 4.3, specifically in the last sentence which states, "All experiments are done on eight A6000 GPUs." However, we consider this cost to be reasonable, given the current trend of recent state-of-the-art models for E2E text spotting to consume large resources for training. For instance, our baselines, SPTSv2 requires 10 A100 GPUs, and UNITS uses 8 A100 GPUs for training.
>
> >*“How does the BAF module affect the reliability of the translated output in cases where errors occur in detection or recognition? Could more error analysis be provided to help understand the effects of the fusion of visual and textual features?”*
>
> We did ablation experiments on BAF module in Table 7, the translation scores gradually improved with the progressive incorporation of feature fusion, which proves the effect of BAF from experiment aspect.
>
> We also provided cases to compare the result with/without BAF in Figure 4. We compared translation results from both UNITS + translation pipeline and our synchronous method under four different scenarios. Event if we use the best text spotter, there still could be recognition error and reading order error. These issues is solved by our synchronous method, also proving the effectiveness of BAF. Except reading order issues mentioned above, about BAF's effectiveness on recognition error,  please see the example below:
> - Figure 4, row 3, column 3: the text in the image was challenging traditional Chinese art fonts, the text spotter mistakenly recognized "雲南" (Yunnan, a province in China) as "餐满" (which can be loosely translated as "full meal").
>
>
> >*“How about the recognition and translation performance of the model in more complex scenarios?”*
>
> We have already compared our layout complexity with the state-of-the-art dataset in the scene TIMT field in the above response. Actually, for current methods, the text in the image is either a single line, like a video caption, or based on a simple scanning reading order, like AnyTrans. Our method demonstrates SOTA  translation performance in such complex layout cases. For more details, please refer to General Response 1. It provides a statistical validation of layout complexity and our translation performance in such complexity.
>
> >*“Can the model be applied to translations in languages other than Chinese and English?”*
>
> Yes, our model is able to be do language extension by applying other language pair's data and re-training.
>
> >*“Can there be a more detailed explanation of the construction of the dataset？”*
>
> Yes, our dataset includes synthesized data and real data. The synthesized data is generated by rendering text content(from WMT22) with multiple fonts on the scene images(selected from COCO). The real data is collected from OCRMT30K, ReCTS, CTW1500 and HierText, and we manually re-labeled the reading order and translation with the help of large language model and prompt. Some details could be found in Section 4.1 DATASETS, and we will share our data generation script along with dataset.

---

### Official Review · Reviewer_iBpM · 2024-11-01

**Soundness:** 2
**Presentation:** 2
**Contribution:** 2
**Rating:** 3
**Confidence:** 5

**Summary:**

This paper proposes a method for synchronous scene text recognition and translation, which introduces a Bridge & Fusion module to integrate text and image modalities. Additionally, the paper collects multiple datasets to construct a scene text image machine translation (TIMT) dataset. The real data includes manually annotated test sets from ReCTS and HierText, while other datasets are annotated using the Qianwen API, followed by manual correction. The synthetic datasets are created based on existing tools. Experimental results demonstrate that the method in this paper outperforms existing text image translation models.

**Strengths:**

1.	A method is proposed that can perform text spotting and text image machine translation, achieving better performance than previous methods.
2.	A Chinese-to-English and English-to-Chinese TIMT dataset is constructed based on multiple existing datasets, which includes annotating real data and synthesizing data using existing tools. This dataset is beneficial to the development of the TIMT field.

**Weaknesses:**

1.	The related work section needs adjustment. This paper mainly focuses on TIMT, yet the authors provide very little introduction to this field in related work, instead offering a large amount of introduction on text spotting.
2.	The paper is somewhat difficult to read and does not provide some key experimental settings. For example, the setting of $\alpha$ in Equation 2 and the size of the Qwen-VL model are not specified. Additionally, all models labeled as 'ours' in the experimental results are bolded, yet they are not the best, which is quite confusing. This result does not align with what the paper claims as SOTA.
3.	The novelty is limited. The Bridge & Fusion module essentially extracts visual features based on the predicted text region's coordinates and then obtains multimodal features through cross-attention. This approach is very common in multimodal machine translation.
4.	The paper claims that its model is end-to-end, however, actually the model still requires autoregressively generating coordinates and recognition output first, and then combines them with the image to autoregressively generate the translation. Therefore, it is not fully end-to-end.
5.	The paper illustrates the issue of incorrect reading order in the pipeline method shown in Figure 1. However, the proposed method does not address this problem but merely provides such training data. Given this type of data, can the text spotting model in the pipeline also solve this issue?

[1] PEIT: Bridging the Modality Gap with Pre-trained Models for End-to-End Image Translation

**Questions:**

1.	What is the size of the Qwen-VL model being compared, and what are the differences in settings with AnyTrans [1]? It would be better if compared with AnyTrans.
2.	What is the specific setting of alpha in Equation 2? Is the model training sensitive to this parameter?
3.	Will the data and code be open source?

[1] AnyTrans: Translate AnyText in the Image with Large Scale Models.

---

> ### Author Response · Authors · 2024-12-02
> **Response to Weakness from Reviewer iBpM**
>
> We greatly appreciate your valuable time and feedback on our paper. Here are our responses to weakness.
>
> >*“The related work section needs adjustment. This paper mainly focuses on TIMT, yet the authors provide very little introduction to this field in related work, instead offering a large amount of introduction on text spotting.”*
>
> Thank you very much for your advice. In the revised paper, we have modified the related work section by streamlining the text spotting content and expanding the coverage of multi-modal translation.
>
> >*“The paper is somewhat difficult to read and does not provide some key experimental settings. For example, the setting of alpha in Equation 2 and the size of the Qwen-VL model are not specified. Additionally, all models labeled as 'ours' in the experimental results are bolded, yet they are not the best, which is quite confusing. This result does not align with what the paper claims as SOTA.”*
>
> - About alpha in Equation 2, as you mentioned in question, the model training is not sensitive to this parameter and you could set it 0.1, 0.5 or 0.9. We have add it in revised paper to enhance the readability and comprehensibility.
>
> - About Qwen-VL model size, the API we utilized is Qwen-vl-max, which is not open-source. Neither the Qwen-VL paper [1] nor its official documentation [2] provides clear information on the model's size. However, we could use the parameters number 9.6B of Qwen-VL in [1] as a reference.
>
> - About bolded not SOTA, it is accurate that we mistakenly used bolded text to emphasize our method rather than the SOTA score. We have rectified this in the revised paper.
>
>
> >*“The novelty is limited. The Bridge & Fusion module essentially extracts visual features based on the predicted text region's coordinates and then obtains multimodal features through cross-attention. This approach is very common in multimodal machine translation.”*
>
> Attention is indeed common in feature fusion. However, different from existing attention[3], we did some special optimization on query in our BAF module by using both position and recognition embedding to search visual feature.  To see more about our novelty, please refer to our General Response 1. It provide details about our contribution on dataset, translation performance and long-tern impact.
>
> >*“The paper claims that its model is end-to-end, however, actually the model still requires autoregressively generating coordinates and recognition output first, and then combines them with the image to autoregressively generate the translation. Therefore, it is not fully end-to-end.”*
>
> Our method is not end-to-end, and we distinguish it from a pipeline because the text spotter and translator are not entirely independent modules, which we emphasize as a schronous method. The categorization in the first column of Tables 2 and 3 did indeed cause some confusion, and we have corrected it and updated the paper accordingly.
>
> >*“The paper illustrates the issue of incorrect reading order in the pipeline method shown in Figure 1. However, the proposed method does not address this problem but merely provides such training data. Given this type of data, can the text spotting model in the pipeline also solve this issue?”*
>
> The annotation of reading order in the data does indeed allow our method to naturally address this issue, but our approach is not entirely data-driven. We employed a combination of ablation experiments to demonstrate our effectiveness:
>
> - In Table 4 of our paper, we selected MCTIT, which scored the highest among the pipeline baselines, for comparison. By eliminating the influence of external OCR tools and even when provided with ground truth OCR results as input, our model's translation scores surpassed MCTIT on the ReCTS dataset, which features the most complex layout (the complexity of which we explained in the "Data" section of General Response 1). This demonstrates that synchronous method can indeed enhance translation capabilities.
>
> - In Table 7 of our paper, the translation scores gradually improved with the progressive incorporation of feature fusion, which proves the effect of BAF. It is noteworthy that the last row of Table 7 “Remove BAF” represents an experiment where the text spotter of our method, which is based on UNITS (the highest-scoring model in the E2E baselines), is used in a pipeline with the translation module. Even if use the state-of-art text spotter, the pipeline's translation score is still lower than our synchronous method.
>
> [1] Qwen-VL: A Versatile Vision-Language Model for Understanding, Localization, Text Reading, and Beyond
>
> [2] Chinese web: https://help.aliyun.com/zh/dashscope/developer-reference/tongyi-qianwen-vl-plus-api
>
> [3] PEIT: Bridging the Modality Gap with Pre-trained Models for End-to-End Image Translation

---

> ### Author Response · Authors · 2024-12-02
> **Response to Questions from Reviewer iBpM**
>
> >*“What is the size of the Qwen-VL model being compared, and what are the differences in settings with AnyTrans [1]? It would be better if compared with AnyTrans.”*
>
> The model size is answered by weakness part. We add comparison with AnyTrans.
> - AnyTrans is also trained on scene TIMT tasks like ours.
> - We test our model on AnyTrans's dataset(Chinese-to-English, also called zh2en) without any fine-tuning.
> - We use the same calculation method as AnyTrans uses in its paper and compare with its highest score(Qwen-vl-max).
>
> Our model achieves comparable BLEU score and higher COMET score compared with AnyTrans.
> | Methods               | AnyTrans: zh2en |         |
> |-----------------------|-----------------|---------|
> |                       | B               | C       |
> | Ours                  | 47.50           |**79.72**|
> | AnyTrans: Qwen-vl-max |**48.70**        | 78.00   |

---

### Official Review · Reviewer_nmXN · 2024-11-05

**Soundness:** 2
**Presentation:** 2
**Contribution:** 2
**Rating:** 3
**Confidence:** 5

**Summary:**

In this work, the authors propose a scene text image machine translation method, which can detect, recognize and translate text in natural scene images. They also create image datasets for both Chinese-to-English and English-to-Chinese image translation tasks.

**Strengths:**

The curated dataset STST800K can be beneficial to the community.

**Weaknesses:**

1. Considering the architecture and pipeline of the proposed model, the novelty of it is limited.
2. The details of casting existing baselines (such as ABCNetv2, SPTSv2 and UNITS) as text spotter and translator are very crucial, but they are absent in the paper.
3. The comparison in Tab. 5 is unfair. The proposed model is pre-trained with data from various sources (such as STST800K and WMT22) and fine-tuned on down-stream datasets, while the Qwen-VL model is not.

**Questions:**

Please address the questions and concerns in the Weaknesses section.

---

> ### Author Response · Authors · 2024-12-02
> **Response to Reviewer nmXN**
>
> We greatly appreciate your valuable time and feedback on our paper. Here are our responses to weakness.
>
> ***
>
> >*“Considering the architecture and pipeline of the proposed model, the novelty of it is limited.”*
>
> Our architecture considers the layout issue(reading order, recognition errors, multi-tasks training) and applies optimization on feature fusion module(BAF) by designing query for attention. Besides, please consider our contributions on dataset, translation performance and long-tern impact on combining large multi-modal model with image preprocessing tasks. Details please see our General Response 1.
>
> >*“The details of casting existing baselines (such as ABCNetv2, SPTSv2 and UNITS) as text spotter and translator are very crucial, but they are absent in the paper.”*
>
> We apologize for any inconvenience caused and have now included clarifications about the casting baselines in Section 3.1 OVERVIEW of the revised paper.
>
> >*“The comparison in Tab. 5 is unfair. The proposed model is pre-trained with data from various sources (such as STST800K and WMT22) and fine-tuned on down-stream datasets, while the Qwen-VL model is not.”*
>
> To keep dataset fairness, we add comparison with AnyTrans[1] in our revised paper:
> - AnyTrans is also trained on scene TIMT tasks like ours.
> - We test our model on AnyTrans's dataset(Chinese-to-English, also called zh2en) without any fine-tuning.
> - We use the same calculation method as AnyTrans uses in its paper and compare with its highest score(Qwen-vl-max).
>
> Our model achieves comparable BLEU score and higher COMET score compared with AnyTrans.
> | Methods               | AnyTrans: zh2en |         |
> |-----------------------|-----------------|---------|
> |                       | B               | C       |
> | Ours                  | 47.50           |**79.72**|
> | AnyTrans: Qwen-vl-max |**48.70**        | 78.00   |
>
> About model size, our model has 2.88B parameters. Qwen-vl-max is is not open-source. Neither the Qwen-VL paper [2] nor its official documentation [3] provides clear information on the model's size. However, we could use the parameters number 9.6B of Qwen-VL in [2] as a reference, which is larger than ours.
>
> [1] AnyTrans: Translate AnyText in the Image with Large Scale Models.
>
> [2] Qwen-VL: A Versatile Vision-Language Model for Understanding, Localization, Text Reading, and Beyond
>
> [3] Chinese web: https://help.aliyun.com/zh/dashscope/developer-reference/tongyi-qianwen-vl-plus-api

---

### Author Response · Authors · 2024-12-02
**General Response 1 - Novelty Limitation**

Reviewers nmXN and iBpM raised concerns about novelty limitation.

>nmXN *“Considering the architecture and pipeline of the proposed model, the novelty of it is limited.”*

>iBpM *“The novelty is limited. The Bridge & Fusion module essentially extracts visual features based on the predicted text region's coordinates and then obtains multimodal features through cross-attention. This approach is very common in multimodal machine translation.”*

***

While we admit that there is a limitation in terms of architecture novelty, we would like to highlight the innovation of our work in the following three aspects.

**1) Data:** All reviewers have acknowledged the potential future benefits that our dataset will bring to the community. Unlike existing datasets in the scene text field, our dataset places a strong emphasis on the significance of reading order in translation tasks. As shown in the table below, our dataset stands out with its greater text richness and layout complexity compared with the latest dataset, AnyTrans[1], which also incorporates reading order. Additionally, it exhibits a remarkable magnitude in the number of images. It could serve as a benchmark for in-depth validation of model structure and training methods in the field of scene text image machine translation.
| Dataset         | Paras per image | Words per para(min/mean/max) | Non-scanning ratio |
|---------------|---------------|----------------------------|------------------|
| AnyTrans: zh2en | 1.38            | 1/1.08/3                     | 0.38%              |
| Ours: ReCTS     | 2.41            | 1/2.25/39                    | 18.03%             |

Paras per image: paragraphs number per image; Words per para: words number per paragraph; Non-scanning ratio: number of paragraphs with non-scanning reading order / total paragraphs number, representing layout complexity.

[1] AnyTrans: Translate AnyText in the Image with Large Scale Models.

**2) Archetecture:** The synchronous architecture aligns with the human thought process of first understanding image content and then translating. Our method, based on existing architectures, specially addresses the layout issue in translation. Besides, it enhances the query of attention in BAF by using both position and recognition embedding to search visual feature. With moderate model parameters, our translation performance outperforms both large multi-modal models (VLM) trained on document tasks and small models trained on specific image translation tasks (pipeline and E2E) as shown in the table below.

| Description | Method               | Parameters | OCRMT30K |         | ReCTS   |         |
|-----------|--------------------|----------|--------|-------|-------|-------|
|             |                      |            | B        | C       | B       | C       |
|             |                      |            |          |         |         |         |
| VLM         | TextMonkey           | 9.7B       | 2.60     | 65.46   | 1.61    | 57.80   |
| VLM         | mPLUG-OWL2           | 8.2B       | 0.90     | 61.06   | 0.59    | 53.61   |
| VLM         | InternVL2-8B         | 8B         | 12.62    | 75.84   | 7.46    | 67.48   |
| VLM         | Qwen2-VL-8B-Instruct | 8B         | 20.75    | 80.73   | 7.80    | 69.61   |
|             |                      |            |          |         |         |         |
| Synchronous | Ours                 | 2.88B      |**35.60** |**84.46**|**28.23**|**75.55**|
|             |                      |            |          |         |         |         |
| Pipeline    | VALHALLA             | 2.6B       | 28.12    | 70.02   | 25.68   | 69.11   |
| Pipeline    | MCTIT                | 1.58B      | 31.07    | 80.93   | 24.42   | 69.79   |
| Pipeline    | E2E-TIT              | 1.37B      | 16.30    | 42.88   | 10.16   | 39.10   |
| Pipeline    | Gated Fusion         | 0.96B      | 29.87    | 75.60   | 24.31   | 66.09   |
| Pipeline    | Selective Attn       | 0.96B      | 26.80    | 70.69   | 21.24   | 66.83   |
|             |                      |            |          |         |         |         |
| E2E         | SWINTS               | 1.77B      | 16.17    | 39.60   | 18.84   | 45.60   |
| E2E         | UNITS                | 1.33B      | 24.06    | 66.34   | 22.45   | 58.31   |
| E2E         | ABCNetv2             | 0.51B      | 2.17     | 33.69   | 4.34    | 16.14   |
| E2E         | SPTSv2               | 0.36B      | 1.90     | 38.32   | 8.50    | 36.80   |

**3) Longterm Impact:** The exploration of synchronous methodologies offers a promising approach in the future for seamlessly integrating multimodal large models with highly precise and specialized preprocessing tasks, which are often challenging for large models. Specifically, the process involves first training a text spotter and then conducting joint fine-tuning with a multimodal large model on tasks related to translation, understanding, and reasoning.

---

### Author Response · Authors · 2024-12-02
**General Response 2 - Compare with Large Model**

According to reviewers iBpM and CjTp's suggestions:
>iBpM *“What is the size of the Qwen-VL model being compared, and what are the differences in settings with AnyTrans [1]? It would be better if compared with AnyTrans.”*

>CjTp *“When comparing with multimodal large models, the analysis lacks a comparison against the most recent large multimodal models, such as mplug-owl (CVPR 24), Monkey (CVPR 24), and InternVL (CVPR 24). It would be beneficial to have the performance results of the large multimodal models after they have been fine-tuned using the corresponding dataset.”*

We supplemented comparisons with more large multi-modal models(VLM), including AnyTrans which is trained on our scene-TIMT tasks, and also the most recent VLMs trained on general document tasks as CjTp recommended. Our model achieves SOTA in a majority of the evaluated scenarios.
- **B**LEU represents the consistency between translations and labels.
- **C**OMET represents the naturalness and intelligibility of translation results.

[1] AnyTrans: Translate AnyText in the Image with Large Scale Models.

***

**(1) AnyTrans**

Our model achieves comparable BLEU score and higher COMET score compared with AnyTrans. To keep dataset fairness, we tested our model directly on AnyTrans's dataset without any fine-tuning and compared the result with the highest score in AnyTrans's paper.
| Methods               | AnyTrans: zh2en |         |
|-----------------------|-----------------|---------|
|                       | B               | C       |
| Ours                  | 47.50           |**79.72**|
| AnyTrans: Qwen-vl-max |**48.70**        | 78.00   |

***

**(2)Most recent large multi-modal models**

- GT Coord + Ours recognition: Input ground truth coordinate to our model and use its recognition as reference input to ask VLM to do visual-textual translation, only testing the translation performance. Our model got apparent higher BLEU score than all VLMs, and only slightly lower than Qwen2-VL in  OCRMT30K - COMET.

- GT Coord + VLM recognition: Do not provide any recognition reference, just draw ground truth coordinate on the image and ask VLM to translate the text within the box, testing the ability to translate directly from image. We achieves SOTA.

- Translate Whole Image: Do not given any information of text regions, just ask VLM to translate the whole image, testing with considerations of both translation and implicit text detection ability. We achieves SOTA.

| GT Coord + Ours recognition |            |          |         |         |         |
|-----------------------------|------------|----------|---------|---------|---------|
| Method                      | Parameters | OCRMT30K |         | ReCTS   |         |
|                             |            | B        | C       | B       | C       |
| Ours                        | 2.88B      |**35.60** | 84.46   |**28.23**|**75.55**|
| Qwen-vl-max                 | -          | 21.92    | 83.74   | -       | -       |
| mPLUG-OWL2                  | 8.2B       | 12.61    | 78.75   | 7.96    | 68.60   |
| InternVL2-8B                | 8B         | 23.06    | 84.22   | 14.47   | 75.16   |
| TextMonkey                  | 9.7B       | 9.69     | 80.85   | 5.78    | 71.00   |
| Qwen2-VL-8B-Instruct        | 8B         | 23.28    |**84.65**| 12.38   | 73.98   |

| GT Coord + VLM recognition |            |          |         |         |         |
|----------------------------|------------|----------|---------|---------|---------|
| Method                     | Parameters | OCRMT30K |         | ReCTS   |         |
|                            |            | B        | C       | B       | C       |
| Ours                       | 2.88B      |**35.60** |**84.46**|**28.23**|**75.55**|
| Qwen-vl-max                | -          | -        | -       | -       | -       |
| mPLUG-OWL2                 | 8.2B       | 0.90     | 61.06   | 0.59    | 53.61   |
| InternVL2-8B               | 8B         | 12.62    | 75.84   | 7.46    | 67.48   |
| TextMonkey                 | 9.7B       | 2.60     | 65.46   | 1.61    | 57.80   |
| Qwen2-VL-8B-Instruct       | 8B         | 20.75    | 80.73   | 7.80    | 69.61   |

| Translate Whole Image |            |          |         |         |         |
|-----------------------|------------|----------|---------|---------|---------|
| Method                | Parameters | OCRMT30K |         | ReCTS   |         |
|                       |            | B        | C       | B       | C       |
| Ours                  | 2.88B      |**23.44** |**55.73**|**24.55**|**58.65**|
| Qwen-vl-max           | -          | -        | -       | -       | -       |
| mPLUG-OWL2            | 8.2B       | 0.36     | 28.84   | 1.01    | 32.71   |
| InternVL2-8B          | 8B         | 3.52     | 49.64   | 2.21    | 50.56   |
| TextMonkey            | 9.7B       | 1.13     | 40.36   | 1.28    |  40.75  |
| Qwen2-VL-8B-Instruct  | 8B         | 2.79     | 50.19   | 1.60    | 51.98   |

---

### Meta-Review · Area_Chair_eMoB · 2024-12-16

**Metareview:**

This paper describes an novel approach to scene text machine translation, addressing the complexities of translating textual content in natural images which possess diverse text and intricate layouts. Unlike traditional methods that separate text spotting and translation tasks, this work proposes a synchronous model that integrates both processes. The authors propose a bridge and fusion module to optimize multi-modal feature utilization and create datasets for Chinese-to-English and English-to-Chinese translation tasks.

The consensus on this paper is clear, with several significant weaknesses identified by reviewers. First, the novelty of the proposed model is minimal, with limited innovation beyond established architectures. Key details regarding the adaptation of existing baselines and the experimental settings are missing, leading to lack of clarity and questions about fairness of experimental comparisons (in particular for the pre-training and fine-tuning procedures). Moreover, while reviewers appreciate the contribution of a new dataset, the proposed approach lacks originality compared to existing multimodal approaches, and gaps in experimental validation raise concerns about the claimed advantages of the proposed framework.

In rebuttal, the authors addressed some of the questions raised by reviewers. However, there remain significant concerns about the experimental validation of the proposed approach and lingering questions about novelty beyond the proposal of a new scene text translation dataset.

Collectively, these shortcomings raised by reviewers outweigh the strengths and indicate that the work is in need of significant revision in form and substance before it can be accepted for publication, and the decision is to Reject.

**Additional Comments On Reviewer Discussion:**

Reviewer comments focused on the questionable novelty of the proposed approach, fairness in some of the comparative evaluations, and gaps in the experimental evaluation. Some of these issues were addressed by the authors in rebuttal, however the limited novelty in the end became the deciding factor.

---

### Decision · Program_Chairs · 2025-01-22

Reject